

# A GCMs-based mathematic model for droughts prediction in the Haihe Basin, China: Multi-GCM Divide-Integration

Dongmei Han[1], Denghua Yan[2], Xinyi Xu[1], Zhongwen Yang[1], Yajing Lu[2]

[1]College of water sciences, Beijing Normal University, No. 19, XinJieKouWai St., HaiDian District, Beijing 100875, P. R. China

[2]Department of Water Resources, China Institute of Water Resources and Hydropower Research. 1-A Fuxing Road, Haidian District, Beijing, 100038, P. R. China

*Correspondence to: Denghua Yan (936282365@qq.com)*

**Abstract.** Recently, the skilful prediction of climate change has drawn high attention from the scientific community. Evidence has been reported the skill of prediction is not satisfactory for the magnitude of inter-annual precipitation and extreme precipitation, and at a smaller spatial scale as well. Based on observational data sets and outputs from the Global Climate Models (GCMs), this study aims at achieving a mathematical model, named multi-GCM divide-integration model (MGDI). The MGDI model is developed by hybridizing finer spatial scale and multi-linear regression model (MLRM) on five state-of-art of GCMs to improve the skills of five GCMs, which is applied to the second level of water resources regionalization in China. It is found that the performance after MGDI model correction has been improved significantly over that of individual GCMs. The errors between observation and simulation after correction (1.6%~4.4%) are within the margin of error (smaller than 5%) and all of the varying trends in each second level of water resources regionalization were same. Furthermore, this study also used the MGDI model to predict the variation of precipitation and droughts at different spatial scale, including second level of water resources regionalization of China and the whole HHB, for the next 40 years. Predictions indicate the climate will gradually change from drying to wetting over the HHB wherein the trend of annual rainfall is 9.3mm/10a. The frequency of drought events will be decreasing as time goes on. The occurrence of mild and severe drought in the Luan River and Jidong Coastal, Tuhai majia River are higher than that in other regions, 9 and 8 respectively. These findings would provide scientific support for current water resources management and future drought-resisting planning of districts in China.

**Keywords.** GCMs, multi-linear regression model (MLRM), multi-GCM divided integration (MGDI),



droughts, Haihe River Basin in China

**1 Introduction**

Under the context of global warming, variable precipitation causes all types of droughts to occur

frequently, widely, continuously and simultaneously (UNISD 2009,Wang 2011, IPCC 2012), including

the agricultural, meteorological and hydrological droughts. Extended meteorological droughts

doubtlessly lead to hydrological droughts, which arise from both the loss of stream flow and reduction

of groundwater level, and in turns to result in agricultural droughts. The severe intensity-duration-

frequency droughts has threatened the sustainable development of social-economy and eco-environment,

and the safety of human life in China.

In recent years, global climate models (GCMs) representing the physical processes of the atmosphere,

ocean and land surface are important tools in simulating the global climate system for climate change

studies (Nair et al, 2015). Topic challenges of GCMs attaching great attention by scholars are generally

the skilful simulation and prediction at a smaller spatial scale (Xin et al., 2012; Chen 2013; Wang et al.,

2014). Some studies usually used statistical downscaling for primary resolution of GCMs to improve

accuracy, but results have generally reported that the performance of the GCMs were not satisfactory for

precipitation at smaller scale over China. Also, previous studies mostly focused on using individual

model or multi-model ensemble mean to assess the skills of GCMs in terms of smaller temporal and

spatial scales, and corresponding results are not successful enough for prediction (Xu et al., 2010a; Wang

et al., 2013; Yan et al., 2014). While plenty of efforts for many years were made, the simulation and

forecastion for climate change is generally not accurate enough because of the coarse resolution of GCMs,

especially for extreme precipitation in a fine scale (Gao et al., 2002; Xu et al., 2010b; Deniz et al., 2012).

However, less research in China focused on the methodology and technique for improving the skill of

GCMs in a view of that there had different application for individual GCM in different spatial and

temporal scale (Han et al., 2015).

Following the above problems on the gap in prediction precipitation at a smaller spatial scale, this paper

proposed a new method to improve the skill of GCMs at different spatial and temporal scale in

consideration of the applicability of GCMs at different spatial and temporal scales. And this paper is

organized as follows. We first (section 2) select the Haihe Basin, China as the study area with the



characters of important strategic position of and high droughts incidence area in China. Drought index selected in this paper is standard precipitation index (SPI) because this index could calculate multi-time

scale. And the correction method for statistical downscaling is MLRM based on the method of statistic. Based on results of previous studies, the paper selected firstly five GCMs from CMIP5 to evaluate their skills in the Haihe Basin, China (section 3.1). After correlation analysis for simulations and observations at different spatial scales, the secondary water resource regionalization in the HHB was selected as minimum appropriate study scale. The multi-GCM divide-integration (MGDI) model is developed by

monthly MLRM calculated by monthly precipitation from five GCMs and observations in each secondary water resource regionalization of the HHB (section 3.2). After verification with the observations, the MGDI model has been further applied in predicting the precipitation (section 3.3) and droughts (section 3.4) for the period of 2011-2050. The objective of this study is to provide a novel methodology for improving skills of GCMs at a fine spatial scale, and to explore how to reduce the

uncertainty inherent in GCMs. This work can provide valuable references for regional sustainable development, flood prevention and drought resistance and property security in the HHB with climate changes.

## 2 Data and methods

### 2.1 Study area

The study area of this research is the Haihe River Basin (HHB) in China, which is a typical region suffering extreme water shortage in China. The HHB is located between 112 ° ~ 120 °E and 35 ° ~ 43 °N, including four secondary regions of water resources: Luan River and Jidong Coastal, North of Haihe, South of Haihe, and Tuhai majia River (Fig.1). Significant changes in temperature and precipitation in the HHB is the main reason of the high occurrence of droughts in recent years. As the region characterized

by low annual runoff, shortage of water resources and tough contradiction between water supply and water demand, the whole environment in the HHB progressively deteriorated.

### 2.2 Data sources

The precipitation datasets used in this study are collected from two types. First, observational daily precipitation data over China are provided by the National Meteorological Information Centre (NMIC)



of China Meteorological Administration. These data are daily gridded rainfall with high-resolution

(0.5°×0.5°) over HHB, and have been used as observed data for the period of 1961-2010.

The other dataset used in this study is the precipitation outputs from five GCMs, including the GFDL-

ESM2M, HADGEM2-ES, IPSL-CM5A-LR, MIROC-ESM-CHEM and NORESM1-Munder under the

RCP2.6 emission scenario from the CMIP5. And the projects are divided into two periods: the simulated

period (1961-2000) and the forecasted period (2011-2050). It needs to be noted that the precipitation

outputs from the GCMs are downscaled and transformed into the same resolution as the observational

dataset, using the methods of statistical downscaling and bilinear interpolation technique (Xu et al., 2014).

In addition, the downscaled data have been further corrected by statistical error based on probability

distribution function (Hagemann et al., 2011; IPS-MIP 2013; Piani et al., 2013).

**2.3 Methods**

In this paper, the spatial and temporal distribution of drought events were discussed. Most drought index

analyse single temporal scale in some period when reflect drought conditions better. But droughts occur

randomly, it is not obviously perfect enough to represent drought used some drought index with a single

time scale. This paper select standard precipitation index (SPI) to analyse drought condition because of

the characters of multiple time scales.

a. Standard precipitation index (SPI)

The standard precipitation index (SPI) proposed by McKee et al. (1993) and Edwards et al. (1997) is

used as a drought index in this study. The SPI represents drought index on different temporal scales

according to more than 30-year rainfall dataset and the formulation is simple and easy to operate.

b. Multi-linear regression model (MLRM)

Despite various improvements in statistical forecast, climate prediction still remains a challenging task.

To reduce the inherent bias that always coexist with the models is inevitable. For this purpose, this study

develop a mathematical model to correct biases in the GCMs outputs.

Here, as follows Eq. (1), multi-linear regression equation has been adopted using statistics correlation

analysis on the platform of Matlab between observed data and predicted rainfall:

$$Y = a + b_1 x_1 + b_2 x_2 + ... + b_k x_k \tag{1}$$

Where  $Y, Y'$--dependent variable, standardized dependent variable;

$x_k, x'_k$--argument, standardized argument;



$b_k, b'_k$ --regression coefficient, standard regression coefficient;

115          a --constant term

Because of the features and dimension of arguments, in study we have used the normalization technique

to standardize, and then re-establish standardized linear regression equation, which is represent in Eq.

(2). The main method that are used to quantify the performance of the developed model are regression

coefficient which can reflect the influence degree of each argument. So the standardize regression

equation is shown as follow:

$$Y' = b'_1 x_1 + b'_2 x_2 + ... + b'_k x_k \qquad (2)$$

Each of the five GCMs outputs and observational data are respectively taken as arguments and dependent

variable to input for above equation. Each regression coefficient calculated from equation are adopted to

correct outputs from GCMs then the feasibility of this method is verified. And the secondary region of

water resources is judged as the study unit since there are higher regression coefficient through

comparing different spatial scales. The MGDI model is constructed based on the method of MLRM.

Thus, the future outputs are predicted using MGDI, wherein future rainfall from five GCMs are taken as

inputs.

### 3. Results and discussion

**3.1 Performance of GCMs**

Based on the downscaled outputs of five GCMs for the period of 1961-2000, this study has evaluated the

predicting skills of the five GCMs over HHB.

The performance of five GCMs are illustrated in Table 2. It can be inferred that the outputs of GCMs are

averagely higher than observations on annual precipitation in HHB and there has less than 5% relative

errors. Thus, all of the GCMs are effective in simulating the distribution of multi-year mean precipitation

over HHB. The trend of inter-annual variability, which is displayed in Fig. 2, shows poor performance

of these GCMs including GFDL-ESM2M, HadGEM2-ES and IPSL-CM5A-LR, and multi-model

ensemble mean as well. And the magnitude of extreme precipitation shows significant degradation in the

prediction of multi-model ensemble mean.

Conclusion, both the performance of individual GCM and multi-model ensemble mean are not skilful to

reflect inaccurately the actual evolution of precipitation, especially for extreme precipitation.



The performance of five GCMs is illustrated in Fig. 3 for multi-year mean at spatial scale. It is found that significant correlation with observed rainfall is displayed. The spatial distribution of rainfall from observation and five GCMs are "more precipitation in southeast and less in northwest" in the HHB as a

whole. And the area with annual average prediction of less than 400mm is smaller, as well as that of 500mm-700mm.

**3.2 Correction of GCMs by monthly MGDI**

Some results can be found that the performance of five GCMs in each month at smaller spatial scales (0.5°×0.5°grid and secondary water resource regionalization) are different. Average correlation

coefficient for each secondary water resources regionalization are into the range of 0.6-0.9 as a whole, but that of the grid scale are not well in correlation between simulation and observation. Hence, the outputs from five GCMs and observations could be further analysed by correlation analysis (Table 3). Therefore, based on above correlation analysis at different spatial scales in the HHB, this study selects the secondary water resources regionalization as the finer scale. So this paper proposed the MGDI method

reduce the bias inherent in GCMs at smaller scale. Monthly MGDI model is constructed to obtain different coefficients of five GCMs and then calculated the corrected outputs from MGDI model. Meanwhile, the MGDI model is used as a tool for predicting the monthly rainfall over the HHB.

The performance of MGDI model in different regions are illustrated in Fig. 4. With a view of Fig. 2, it can be inferred the performance of MGDI model has been significantly improved over that of individual

GCMs as a whole.

The performance of MGDI model in multi-year mean at the finer scale is presented in Table 4. One of the interesting thing that can be seen here is that after the correction, the skill of MGDI model increased for each secondary region of water resources. The predicted trends of MGDI model are closer to those of observations, values of which are higher than that of each GCM. It is worth mentioning that the margin

of error presents a significant decrease and the uncertain projection of GCMs reduces enormously.

In view of the above, the discussion yielded to important points: (1) After the correction of the GCMs outputs, namely MGDI model, the performance of prediction in each month at a smaller spatial scale of the HHB has improved significantly. So, for the better forecast, it is essential and important to correct the GCMs in most cases. (2) Developing the MGDI model, which is statistical model, is to significantly

decrease the uncertainty of the GCMs.





### 3.3 Forecasting precipitation for 2011-2050

Based on the future outputs of five GCMs under the RCP2.6 emission scenario, this study adopts the MGDI model to calculate monthly precipitation over HHB and each secondary region of water resources. Annual precipitation over HHB is on the rise from 2011 to 2050 and the inter-annual variation of precipitation changes significantly. The maximum of annual precipitation will occur in 2040s (728.6mm) and it will be 1.8 times more than the minimum value. Multi-year average precipitation is 540.7mm in the future 40-year, which increase by 2.6% than that in the past 50-year. And as decades goes on, annual precipitation will increase gradually and the tendency value reach up to 9.3mm/10a (Table 5).

It can be noted in Fig.6 that annual precipitation in every secondary region of water resources has been increasing from 2011 to 2050 as a whole, especially in Tuhai majia River wherein the slope of precipitation trend is greatest. Also, the drastic variation of annual precipitation in Tuhai majia River is more significant than those in other basins, so it's obvious that the extreme climate events might occur more frequently. After 2025, annual precipitation in south of Haihe starts to increase significantly, while the precipitation in the basins, such as Luan River and Jidong Coastal, North of Haihe, varies irregularly. The multi-year mean precipitation in Luan River and Jidong Coastal is highest over HHB. Average annual precipitation in every secondary region of water resources reaches the highest value in 2040 that contributes much to the total precipitation of the HHB. (Table 6).

### 3.4 Forecasting droughts for 2011-2050

It is presented in Fig.7 there is an increasing tendency in annual SPI value over HHB from 2011 to 2050 and the climate over HHB tends to be wetter as a whole. The droughts degree gradually reduces. Significant change of SPI value after 2025 is mainly caused by uneven distribution of precipitation. D-1, D-2 droughts (D-1, D-2 represent moderate droughts and severe droughts, respectively) occur six times over HHB, in which D-2 droughts occurs twice.

Increasing slightly of the annual SPI values in each secondary region of water resources indicates that the degree of droughts reduces. Significant variation of annual SPI value symbolizes that alternate floods/droughts will occur frequently over most parts of the HHB, especially Luan River and Jidong Coastal, Tuhai majia River (Fig.8). From Table 8, it can be inferred that droughts in secondary regions of water resources occurs frequently in 2010s. The occurrence of D-1 and D-2 droughts show the values in the Luan River and Jidong Coastal, Tuhai majia River are higher than that in other regions. It is worth



mentioning that alternate floods/droughts will occur in every secondary region of water resources and
the number of D-2 droughts year is increasing inevitably.

The spatial distribution of droughts is presented in Fig.9. There has been probably increasing tendency
from southwest to northeast over the HHB from 2011 to 2050. Distribution of D-1 droughts is identical
to those of total droughts in each secondary region of water resources. It is also seen from the distribution

of D-2 droughts that the southeast region is more prone to occur than the rest of the HHB, but the
difference of droughts occurrence is not obvious. The D-3 droughts, namely extreme droughts, occurs
only once in North of Haihe.

The spatial distribution of droughts in all of secondary regions of water resources are various in different
years. On the average, the occurrence of droughts in the northeast region is slightly higher. In 2010s, the

frequency of droughts increases gradually from western to eastern as a whole. In the next decade, in
addition to the South of Haihe, the frequency of droughts presents an increasing trend from south to north.
In 2030s, the spatial distribution of droughts is irregular that the frequency of droughts occurred in the
southeast of the HHB is slightly less. And in 2040s, the frequency of droughts in most parts of the HHB
are basically the same except that the frequency of droughts in the South of Haihe is just less than other

regions once (Fig.10).

## 4. Summary and conclusion

Because of the effects on the change of climate, land use, ecological environment and other economic
factors in HHB, extreme climate events (such as droughts and floods) occur with increasing frequencies
in recent years. It is seen that annual precipitation decreased rapidly and the area of droughts increased

over the HHB from 1961 to 2010. The area wherein the annual precipitation was below 500mm was
expanding, while the area of precipitation between 600mm and 800mm was reduced. And the belt of
annual precipitation moved southwards in recent 50 years.

With the increasing use of the GCMs, the skill of prediction has relatively improved. But at a smaller
spatial scale, the accuracy of prediction decreases as the models are downscaled to higher resolution. The

aim of this study is to improve the skill of GCMs in predicting the variation of precipitation and droughts
and achieve a mathematical method for correcting bias in GCMs at a smaller spatial scale.



It is seen that before the MGDI model is constructed, the skill of the GCMs is significantly correlated with observations for multi-year mean precipitation over HHB. But the five GCMs performance is not satisfactory in predicting the magnitude of inter-annual precipitation and extreme precipitation. After

correlation analysis, the value of correlation in the secondary regions of water resources are significantly better than those in the grid level. So this study selects the secondary region of water resources as a fine spatial scale. Meanwhile, a new mathematical method, namely MGDI model is proposed, can decrease errors and improve the skill of the GCMs, significantly.

After the MGDI model, this study estimates the revolution of precipitation and droughts at the all-HHB

level and at the secondary regions of water resources level also in the next 40 years. It is found that there is an increasing tendency of annual precipitation in every secondary region of water resources and in the HHB as a whole. The forecasting of the droughts is judged using the MGDI model and it is noted that both the frequency of droughts and degrees of droughts declines in every secondary region of water resources, as the increasing precipitation in those regions contributes much to the results. Droughts occurs

more frequently in the northeast of the HHB than the rest parts of the HHB, and the degrees of droughts are mainly moderate and severe. It is also found, even though the frequency of extreme droughts will reduce in the HHB as a whole, the southwest of the HHB is still the region of high-incidence droughts in the future 40 years.

**Acknowledgements.**

This work was supported by the General Program of the National Natural Science Foundation of China (Grant No. 51279207 and 51409266) and the National Key Technology R&D Program (Grant No. 2013BAB05B04).

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

**Table 1: Degree and description of droughts based on SPI value.**

| SPI | $(-\infty, -2]$ | $(-2, -1.5]$ | $(-1.5, -1]$ | $(-1, +\infty)$ |
|---|---|---|---|---|
| Degree | 3 | 2 | 1 | 0 |
| Description | Extreme droughts | severe droughts | moderate droughts | normal |

**Table 2: The performance of GCMs for multi-year average precipitation over HHB during 1961-2000**

| Indicator name | Observations | GFDL-ESM2M | HadGEM2-ES | IPSL-CM5A-LR | MIROC-ESM-CHEM | NorESM1-M |
|---|---|---|---|---|---|---|
| Multi-year average precipitation(mm) | 535.4 | 521.8 | 524.9 | 514.4 | 519.4 | 518.8 |
| Tendency(mm/10a) | -20.9 | -6.4 | -6.8 | -2.0 | 9.2 | 1.3 |


**Table 3: Correlation coefficient between observation and simulation of monthly precipitation in different secondary water resources regionalization during 1961-2000**

| Division | GFDL-ESM2M | HadGEM-2ES | IPSL-CM5A-LR | MIROC-ESM-CHEM | NorESM1-M |
|---|---|---|---|---|---|
| Luan River and Jidong Coastal | 0.81 | 0.77 | 0.81 | 0.77 | 0.83 |
| North of Haihe | 0.79 | 0.79 | 0.78 | 0.78 | 0.82 |
| South of Haihe | 0.75 | 0.73 | 0.73 | 0.74 | 0.79 |
| Tuhai majia River | 0.73 | 0.68 | 0.67 | 0.70 | 0.79 |





**Table 4: The performance of MGDI model for multi-year average precipitation in each secondary water resources regionalization during 1961-2000.**

| Region | Name | Observations | Correction | error |
|---|---|---|---|---|
| Haihe Basin | annual precipitation(mm) | 535.4 | 519.3 | 3% |
| | trend(mm/10a) | -20.9 | -19.3 | Same trend |
| Luan River and Jidong Coastal | annual precipitation(mm) | 567.5 | 555.0 | 2.2% |
| | trend(mm/10a) | -8.2 | -9.5 | Same trend |
| North of Haihe | annual precipitation(mm) | 503.5 | 495.4 | 1.6% |
| | trend(mm/10a) | -7.4 | -9.7 | Same trend |
| South of Haihe | annual precipitation(mm) | 536.8 | 533.2 | 4.4% |
| | trend(mm/10a) | -25.2 | -23.3 | Same trend |
| Tuhai majia River | annual precipitation(mm) | 576.4 | 566.1 | 1.8% |
| | trend(mm/10a) | -35.1 | -32.5 | Same trend |


**Table 5: Chorological variation in terms of annual precipitation over HHB. The colour arrows indicate the level of magnitude of annual mean precipitation in each eras.**

| decade | | 2010s | | 2020s | | 2030s | | 2040s | trend rate |
|---|---|---|---|---|---|---|---|---|---|
| annual precipitation | ⬇ | 532.3 | ⬇ | 535.7 | ⇨ | 543.6 | ⬆ | 560.6 | 9.3 |

**Table 6: Variation in annual precipitation of each secondary water resources regionalization in different eras during 2011-2050. The colour arrows indicate the level of magnitude of annual mean precipitation in each eras.**

| decade | | 2010s | | 2020s | | 2030s | | 2040s | trend | trend rate(mm/10a) |
|---|---|---|---|---|---|---|---|---|---|---|
| Luan River and Jidong Coastal | ⬇ | 580.6 | ⬇ | 585.7 | ⬇ | 590.3 | ⬆ | 611.6 | ▬ ▬ ▬ ■ | 9.8 |
| North of Haihe | ⬇ | 525.0 | ⬇ | 530.7 | ⬇ | 524.4 | ⬆ | 547.3 | ▬ ▬ ■ | 6 |
| South of Haihe | ⬇ | 520.2 | ⬇ | 515.1 | ⇨ | 534.9 | ⬆ | 546.6 | ▬ ■ ■ | 9.9 |
| Tuhai majia River | ⬇ | 526.0 | ⬆ | 560.5 | ⇨ | 554.8 | ⬆ | 574.0 | ■ ■ ■ | 13.8 |



**Table 7: Number of different degrees droughts for the individual eras in secondary water resources regionalization during 2011-2050.**

| Eras and Degree | Luan River and Jidong Coastal | North of Haihe | South of Haihe | Tuhai majia River |
|---|---|---|---|---|
| 2010s | 3 | 2 | 1 | 4 |
| 2020s | 2 | 0 | 1 | 1 |
| 2030s | 2 | 2 | 3 | 1 |
| 2040s | 2 | 2 | 1 | 2 |
| Droughts | 9 | 6 | 6 | 8 |
| D-1 | 7 | 3 | 4 | 5 |
| D-2 | 2 | 2 | 2 | 3 |
| D-3 | 0 | 1 | 0 | 0 |

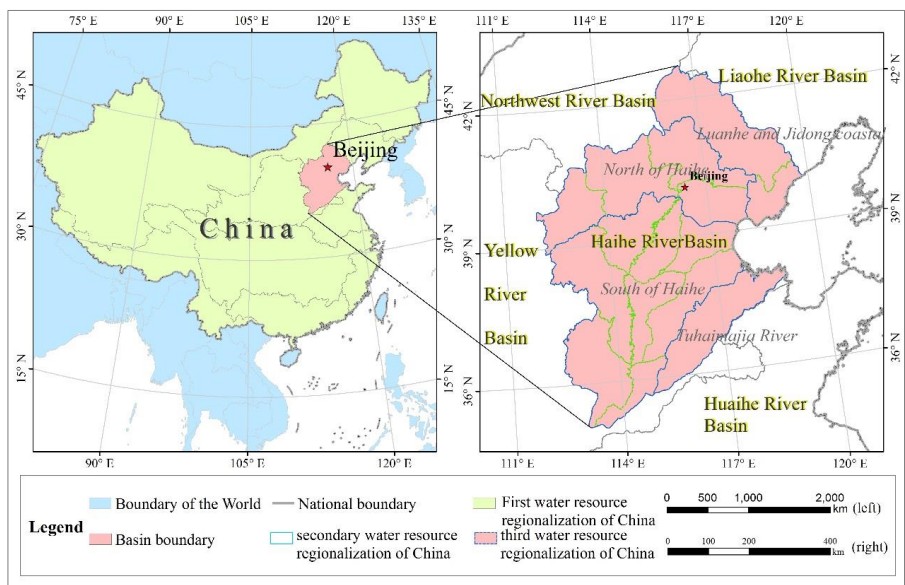

**Figure 1: Space location map of the HHB in China including four secondary water resources regionalization (divided by blue lines), and fifteen third water resources regionalization (divided by green lines)**




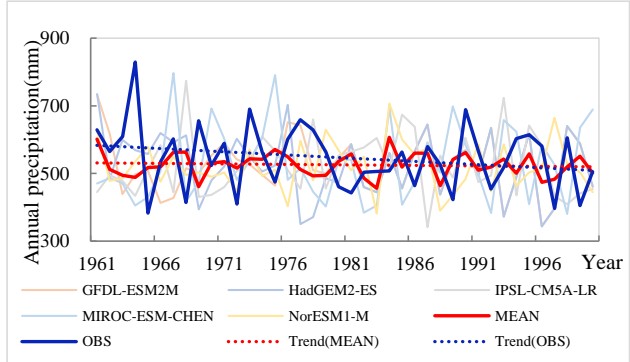

**Figure 2: The performance of five GCMs and multi-model ensembles mean over HHB during 1961-2000.**
**(The blue line represents the variation curve of observational values, the red line represents the changing**
**curve of multi-model ensemble mean, and the blur lines represent the variation curve of the outputs from**
**five GCMs.)**

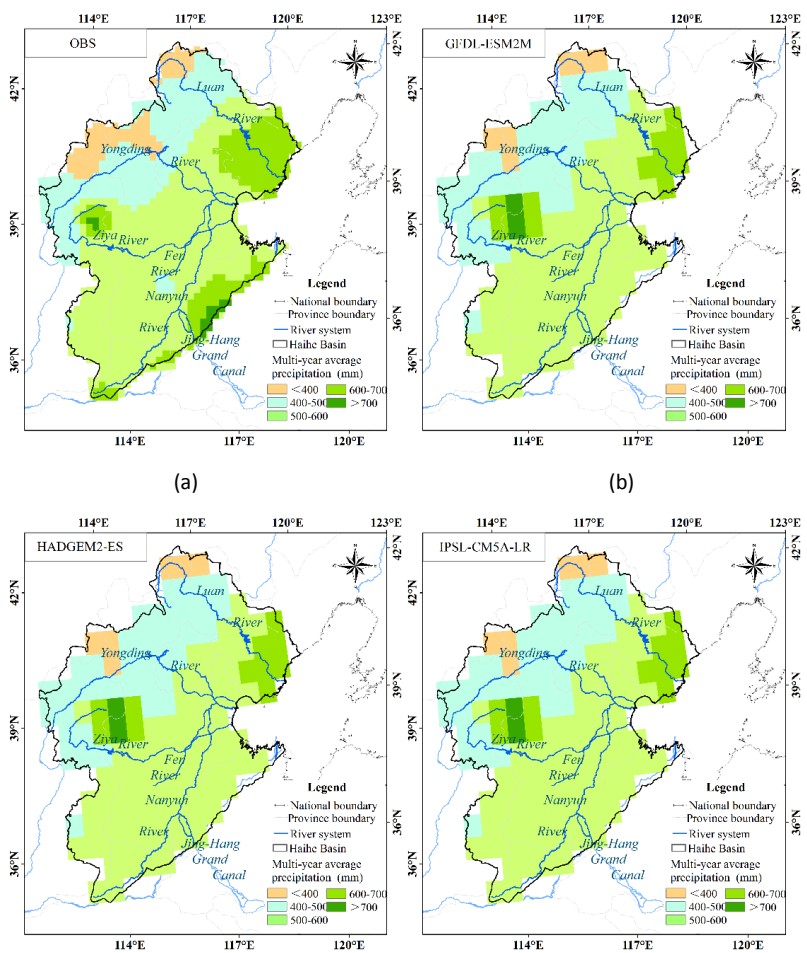




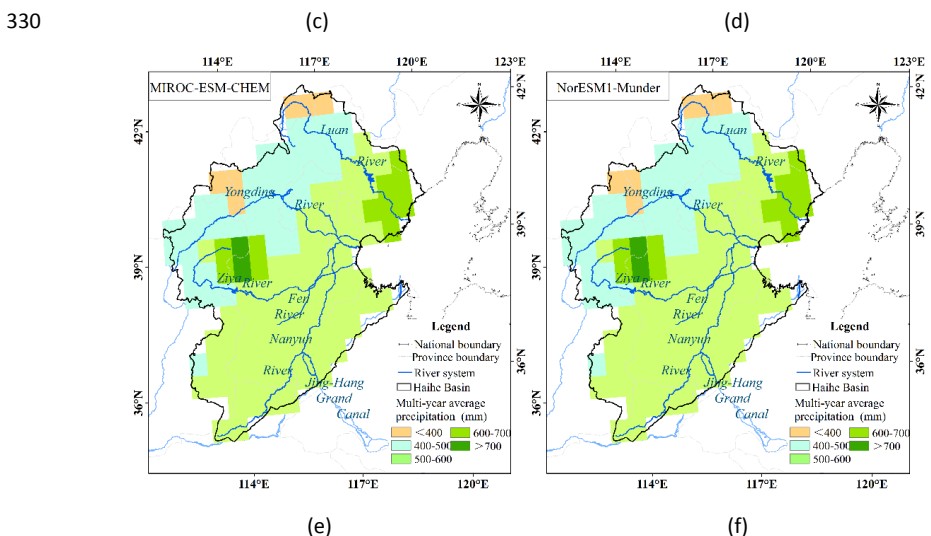

**Figure 3: Comparison of spatial distribution of multi-year average precipitation between simulations from five GCMs and observations.**


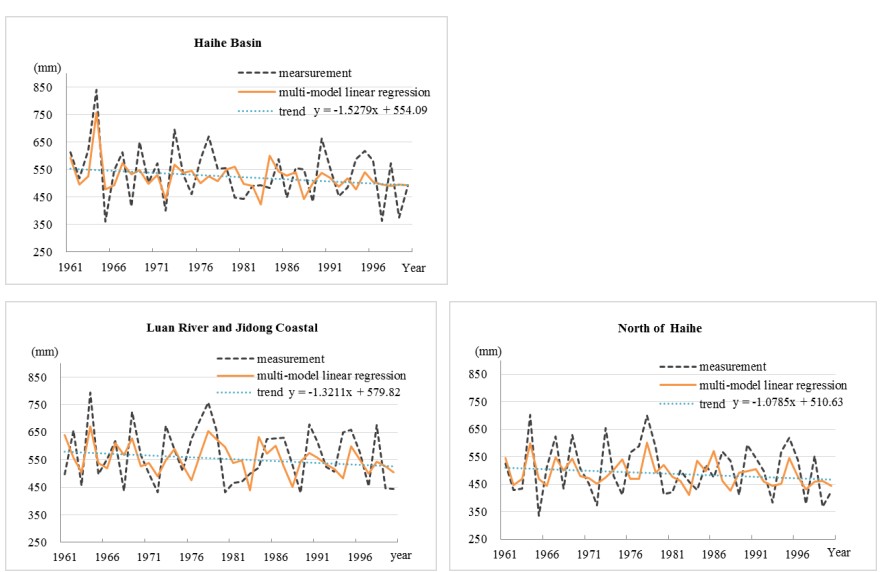



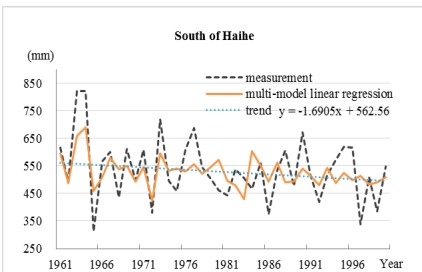
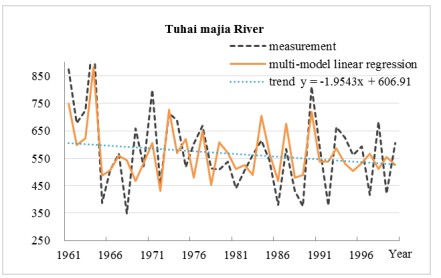

**Figure 4: Comparison of measured and corrected annual precipitation in each secondary water resources**

**regionalization during 1961-2000.**

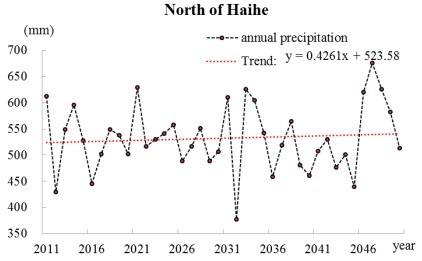

**Figure 5: Inter-annual variation in terms of annual precipitation over HHB. The numbers on the y-coordinate**
**and the x-coordinate indicate number of annual precipitation and years, respectively. And the red line**
**represents the trend line of precipitation.**

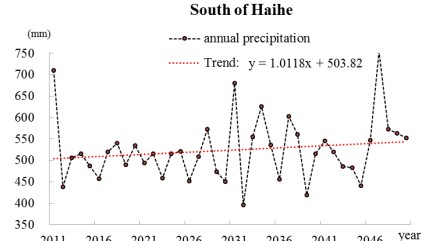
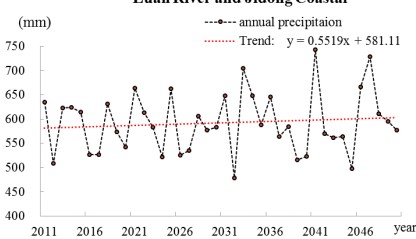
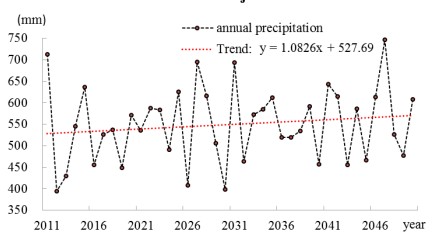





**Figure 6: Variation of annual precipitation in each secondary water resources regionalization during 2011-2050. The numbers on the y-coordinate and the x-coordinate indicate number of annual precipitation and years, respectively. And the red line represents the trend line of precipitation.**


**Figure 7: Variation in annual SPI values over HHB during 2011-2050. The numbers on the y-coordinate and the x-coordinate indicate degree of droughts and years, respectively. And the red line represents the trend line of SPI values.**


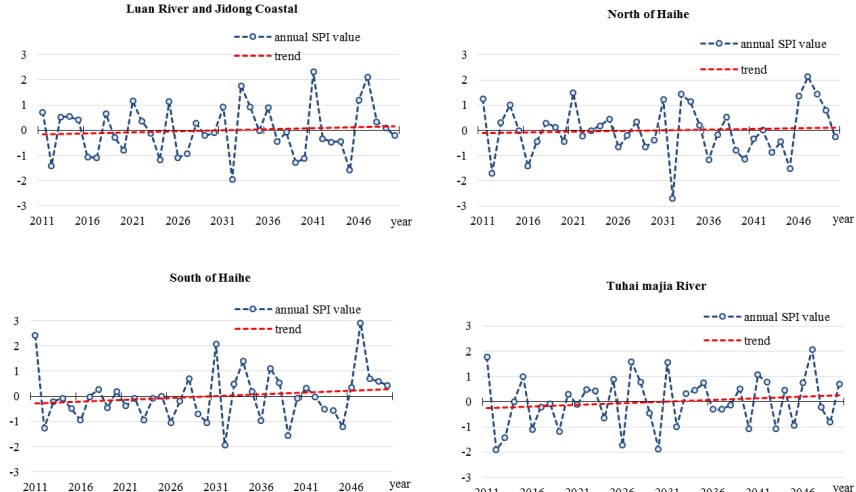

**Figure 8: Variation in annual SPI values in secondary water resources regionalization of China during 2011-2050.**






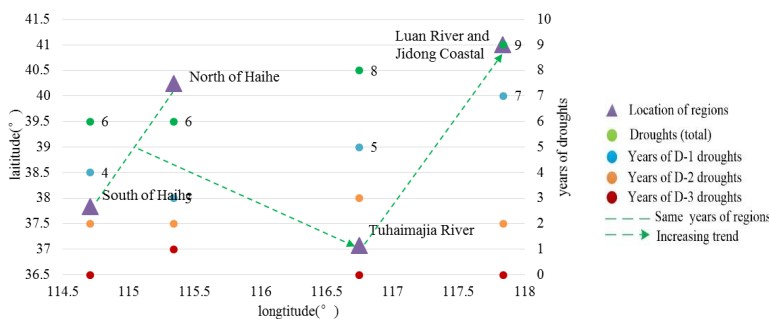

**Figure 9: Spatial variation of different degrees droughts in HHB in future 40 years (The x-axis symbols the longitude location of regions. The left y-axis symbols the latitude location of regions and the right y-axis symbols the years of different degree droughts. The triangle corresponds to the x-axis and left y-axis and the colourful circles correspond to the x-axis and right y-axis. The green lines shows the spatial trend of total droughts.)**


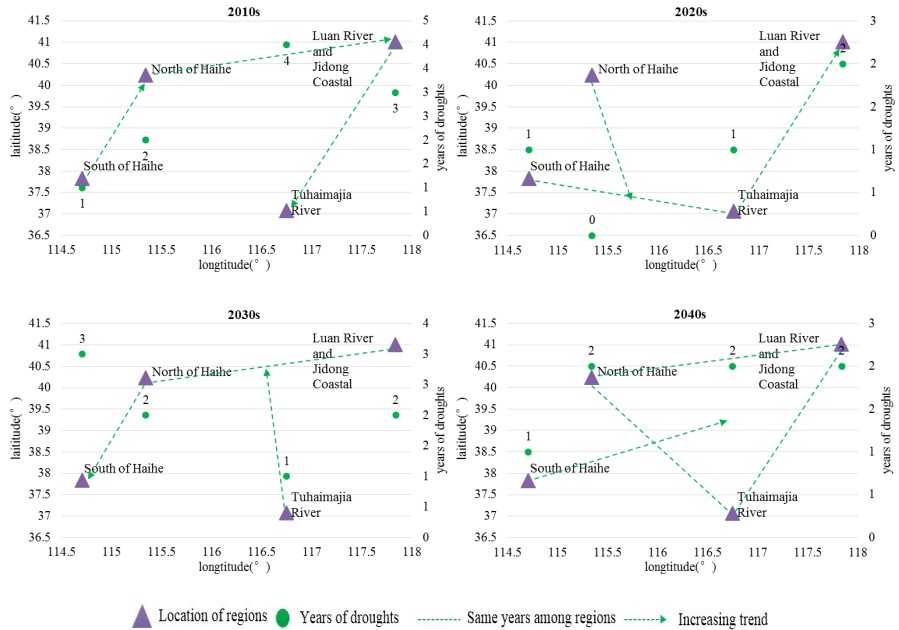


**Figure 10: Spatial variation of droughts in HHB in future different eras (The x-axis and the left y-axis symbol the location of regions which correspond to triangles. And the right y-axis symbols years of different degree droughts which marked in colourful circles. The green lines shows the spatial trend of total droughts.)**