# Peer review of "A GCMs-based mathematic model for droughts prediction in the Haihe Basin, China: Multi-GCM Divide-Integration"

_Natural Hazards and Earth System Sciences, 2016_

## Short Comment (SC1) · 21 Jun 2016

Dear Han, D. doi: 10.5194/nhess-2016-150 Title: A GCMs-based mathematic model for droughts prediction in the Haihe Basin, China: Multi-GCM Divide-Integration

Recommendation: the paper is probably publishable, but should be reviewed again in revised form before it is accepted as the subject is interesting.

Additional comments: In the present work, the authors introduce and propose a new method of GCM-based to improve the skill of prediction of global climate models (GCMs). The new methods is built by multi-linear regression model. Through the results you showed, this paper introduce more about the innovations of MGDI model

with significant improvement for performance of GCMs. And a more important point is the study area selected in this paper is Haihe basin which is a focal research region with important strategic position in China, so the results obtained provided scientific and technological support for region for regional adaptation and mitigation strategies to address climate change. Other minor points are: -the overall quality of the English language is improved by native language and some sentence are not correct. -the literature view is poor, this paper should be supported by references.

Once the above concerns are fully addressed, the manuscript can be accepted for the publication in this journal.

---

## Author Comment (AC1) · 29 Jun 2016

Thank you very much for your comments on 21 Jun 2016 with which you sent to interactive discussion on our paper with the doi number 10.5194/nhess-2016-150. We also wish to take this opportunity to thank the reviewer for her constructive comments and valuable recommendations. We have carefully revised the manuscript according to reviewer's suggestion.

Our responses to several comments are listed below:

Comments 1: In the present work, the authors introduce and propose a new method of GCM-based to improve the skill of prediction of global climate models(GCMs). The new

[Figure]

methods is built by multi-linear regression model. Through the results you showed, this paper introduce more about innovations of the MGDI model with significant improvement for performance of GCMs. And a more important point is the study area selected in this paper is Haihe basin which is a focal research region with important strategic position in China, so the results obtained provided scientific and technological support for region for regional adaptation and mitigation strategies to address climate change. Reply 1: First extremely thanks the reviewer's approval and so in-depth analysis of the manuscript. The manuscript proposed the new approach which effectively improve the skill of GCMs simulation.It is important to strengthen the management and analyze the risks of future extreme events (drougths, floods and extreme weather events), especially in national key regions.

Comments 2: The overall quality of the English language is improved by native language and some sentence are not correct. Reply 2: The language and some sentence will be further modified according to the native speaker revision.

Comments 3: The literature view is poor, this paper should be supported by references. Reply 3: Thank you for your valuable advice and we consider adding to several reference in introduction part and in method part to enrich the content and provide the foundations for this manuscript.

---

## Short Comment (SC2) · 6 Jul 2016

Dear Han, DOI: 10.5194/nhess-2016-150 Manuscript entitled "A GCMs-based mathematic model for droughts prediction in the Haihe Basin, China: Multi-GCM Divide-Integration".

Firstly, it is worthy to mention that I have studied this subject recently and have read many literature on development and application of GCMs. This study proposed a new procedure for improving the skills of GCMs, I think it is of particular value in application, and the procedure has the characteristics of simple-convenient operation and universal application over the previous methods. As seen from the results that the skill of simulation based on new approach improved significantly. And I think the conception

or the idea on establishment of the new approach is actually good, that is taking full advantage of simulation of individual GCM on suitable spatial-temporal scales. Have a few comments: 1. The paper should detail this new approach compared with other methods, and explains why that is better or different from other methods. 2. The literature review part of the manuscript is limited, and missing some papers written in this topic. I strongly advice authors to refer the existing literature for further understanding. 3. Some sentence are necessary to correct for clear-exact understanding.

So, viewed as a whole, I suggest this manuscript be accepted and published for its certain reference value before the above comments are fully revised.

---

## Referee Comment (RC1) · Anonymous Referee #1 · 15 Jul 2016

The paper deals with an application of a standard statistical downscaling technique, namely multi-linear regression, to some CMIP5 runs over the Chinese region of the Haihe Basin, using an observational high resolution precipitation analysis. Although the topic of the paper may be of interest and publishable, the overall level of the paper is too poor for publication. The English is poor to the point that understanding what is written by the authors is more an imagination than a reading exercise. There is no point making a list of the English mistakes: the language should be improved by someone who has a sufficient English knowledge. As for the scientific contents, I can see at least two major concerns. The first problem is that the authors do not apply the statistical downscaling method in cross validation, that is the data used to

calibrate the multi-linear regression scheme are the same used to validate the results. As a result, the evaluation of the skill of the downscaling is affected by over-fitting. The authors should divide the observational and present day model data (1960 to 2010) into two parts: one used to calibrate and the other to validate the downscaling scheme. Possibly they should re-do the exercise inverting the two data-sets and see if the results obtained in the two cases are consistent. This should proof the robustness of the results. The second problem is that the authors probably reassured by the high correlation values of the downscaling results, describe the results obtained with great confidence, comparing the climate predictions obtained for different decades, and indicating the exact number of droughts that will occur in each decade together with their intensity, using as drought index the SPI. Now, even if the downscaling scheme output skill were correctly evaluated, the precipitation trend over the Haihe Basin would be known within 40 to 50% of relative error (I have computed this numbers starting from the Table 4). This does not allow to express the prediction by a number. The authors are actually using an ensemble of predictions, which is correct, and they should produce a probabilistic prediction, not a deterministic one. Once the authors have completely rewritten the manuscript they should also chose a more appropriate title for it. Although math is always involved in all climate predictions, the present title would suggest that the topic is a description of results obtained by applying a theoretical approach, which is absolutely not the case for this paper.
* * *

---

## Referee Comment (RC2) · Anonymous Referee #2 · 19 Jul 2016

Review of the Manuscript: 'A GCMs-based mathematic model for droughts prediction in the Haihe Basin, China: Multi-GCM Divide-Integration'

by Dongmei Han, Denghua Yan2, Xinyi Xu, Zhongwen Yang, Yajing Lu

Not acceptable for publication in this form

The subject of the paper is quite relevant and potentially publishable, trying to combine independent GCM simulations of the SPI at the regional scale in order to get more skillful hindcasts. Then the authors apply the hybrid calibrated model to produce predictions of precipitation and number of droughts in the next future decades.

Main Comments:

1) Despite the potential interest of the manuscript, the English writing is quite poor and confuse with the presence of many syntactic grammar mistakes which puts serious problems concerning the correct understanding of the author's messages. In a future resubmission, it is suggested that authors ask for help from a speaking English native.

2) Beyond that, there are significant methodological errors thus making some of the authors' conclusions quite useless, in particular in what respects the decadal predictions. Moreover, there are relevant details in the regression model which remain unclear or absent. For instance, the regression is done point by point or for a whole area? The regression is done on a monthly or annual basis?

3) The simulation score evaluation from the MLRM must be done in cross validation mode (e.g. leave one year out strategy). That will reduce the strong over fitting effect by positively biasing the correlation skill.

4) The method is not detailed enough with some crucial points remaining unclear. For instance, does the linear regression 2 is performed in a point by point fashion with results summed over each basin or the converse is done with regression being computed for the regional SPI as a whole?

5) Throughout the paper, there is no statistical significance study present at all, both in calibration and validation period. Giving the overfitting effect, the regression coefficients may be affected by substantial levels of error. The confidence intervals for the issuing simulations (in the calibration period) and forecasts (in the future period) should be added. After applying that and looking for what is really statistically significant, maybe many of the author's conclusions regarding the SPI decadal trends and the expected number of droughts in the coming decades will become not plausible.

Small Comments

1) The 'Divide-Integration' model's attribute is not appropriate. 2) Authors shall mention the SPI references when it is referred for the first time in the paper. 3) Concerning the

monthly MGDI model. Regression coefficients are computed for each month, rather than for the annual values? If yes, the over-fitting effect is still more drastic than when doing regression on a annual basis. 4) The style of references section is not the proper one. The body of individual references is not indented which makes difficult to distinguish each reference. 5) From the text, in line 150, authors refer to average correlation coefficient for each secondary basin. An average of correlation coefficients over an area has no sense. First, both observations and simulations are summed over the area, then the correlation coefficient is computed. 6) The Fig. 3 is not much informative, being somehow redundant (Figs 3c,d,e,f are equal). Climatological biases of the annual precipitation are preferable. It would be rather worth to give biases of the model ensemble mean.